# Evaluating the Antihyperalgesic Potential of Sildenafil–Metformin Combination and Its Impact on Biochemical Markers in Alloxan-Induced Diabetic Neuropathy in Rats

**DOI:** 10.3390/ph17060783

**Published:** 2024-06-14

**Authors:** Ciprian Pușcașu, Simona Negreș, Cristina Elena Zbârcea, Anca Ungurianu, Emil Ștefănescu, Nicoleta Mirela Blebea, Cornel Chiriță

**Affiliations:** 1Faculty of Pharmacy, “Carol Davila” University of Medicine and Pharmacy, Traian Vuia 6, 020956 Bucharest, Romania; ciprian.puscasu@umfcd.ro (C.P.); simona.negres@umfcd.ro (S.N.); anca.ungurianu@umfcd.ro (A.U.); emil.stefanescu@umfcd.ro (E.Ș.); cornel.chirita@umfcd.ro (C.C.); 2Faculty of Pharmacy, “Ovidius” University of Constanța, Căpitan Aviator Al. Şerbănescu 6, 900470 Constanța, Romania; nicoleta.blebea@gmail.com

**Keywords:** diabetic neuropathy, sildenafil, metformin, antihyperalgesic, IL-6, TNF-α

## Abstract

(1) Background: Globally, about 600 million people are afflicted with diabetes, and one of its most prevalent complications is neuropathy, a debilitating condition. At the present time, the exploration of novel therapies for alleviating diabetic-neuropathy-associated pain is genuinely captivating, considering that current therapeutic options are characterized by poor efficacy and significant risk of side effects. In the current research, we evaluated the antihyperalgesic effect the sildenafil (phosphodiesterase-5 inhibitor)–metformin (antihyperglycemic agent) combination and its impact on biochemical markers in alloxan-induced diabetic neuropathy in rats. (2) Methods: This study involved a cohort of 70 diabetic rats and 10 non-diabetic rats. Diabetic neuropathy was induced by a single dose of 130 mg/kg alloxan. The rats were submitted to thermal stimulus test using a hot–cold plate and to tactile stimulus test using von Frey filaments. Moreover, at the end of the experiment, the animals were sacrificed and their brains and livers were collected to investigate the impact of this combination on TNF-α, IL-6, nitrites and thiols levels. (3) Results: The results demonstrated that all sildenafil–metformin combinations decreased the pain sensitivity in the von Frey test, hot plate test and cold plate test. Furthermore, alterations in nitrites and thiols concentrations and pro-inflammatory cytokines (specifically TNF-α and IL-6) were noted following a 15-day regimen of various sildenafil–metformin combinations. (4) Conclusions: The combination of sildenafil and metformin has a synergistic effect on alleviating pain in alloxan-induced diabetic neuropathy rats. Additionally, the combination effectively decreased inflammation, inhibited the rise in NOS activity, and provided protection against glutathione depletion.

## 1. Introduction

Diabetes and its complications are growing challenges for healthcare systems around the world. Diabetes affects about 600 million people globally, and by 2045, that figure is expected to rise to 700 million [1].

Neuropathy is a frequently occurring complication of diabetes. This pathology is characterized by high morbidity and poor quality of life [2]. The prevalence rates of diabetic neuropathy (DN) vary depending on the region and the accuracy of prevalence may be affected by differences in diagnostic techniques and criteria. Moreover, the duration of diabetes also plays a crucial role in the progression of DN [3]. Typically, the incidence of neuropathy is higher in people with type 2 diabetes [4]. In addition, another recent study including 25,710 type 2 diabetic patients from China revealed a percentage of 57.2% DN among them [5]. Moreover, a study including 3000 diabetics from 16 countries reported a 28% prevalence of DN [6]. Aldana et al. estimated a prevalence of DN of 46.5% in a study including eight countries from Latin America [7]. In Europe, the prevalence of this pathology ranges from 6% to 34% among diabetics [8].

The mechanisms underlying the onset and progression of DN remain poorly understood, resulting in undertreatment [9]. The present literature on the pathophysiology of DN demonstrates the involvement of oxidative stress, the polyol and hexosamine pathways, proinflammatory cytokines, the sodium and calcium channels, microvascular alterations, and insulin signaling pathways [4,10]. 

Over time, numerous preclinical and clinical studies have provided evidence that systemic inflammation is involved in the pathogenesis of DN. Both TNF-α (tumor necrosis factor) and IL-6 (interleukin-6) are pro-inflammatory cytokines whose production is increased in patients with DN [4,11]. Due to chronic hyperglycemia, nitric oxide synthase (NOS) depletion occurs, which consequently leads to free radical generation and an increase in the production of nitric oxide (NO) [12]. By altering the injured peripheral axons, NO plays an important role in the development of DN [4]. On the other hand, oxidative stress mediated by free radicals also has implications in the progression of diabetes and its complications and both types of diabetes demonstrated low levels of protein thiols, with these decreases being attributed to metabolic and inflammatory changes [13,14]. 

Currently, besides the high risk of side effects, therapeutic strategies have limited success in reducing the pain caused by DN [15]. Considering these, the discovery of new treatments is of real interest. Sildenafil, a PDE5 (phosphodiesterase 5) inhibitor, demonstrated its ability to reduce pain from neuropathy by increasing cGMP (cyclic guanosine monophosphate), which consequently enhances the production of γ-aminobutyric acid (GABA) [16]. Furthermore, chronic hyperglycemia produces an increase in the level of PDE5, and sildenafil, by inhibiting this enzyme, contributes to an enhancement in the symptoms of DN [17]. 

On the other hand, metformin, besides its antihyperglycemic effect, has also demonstrated antinociception action in several studies. Thus, it alleviates pain through multiple mechanisms, including AMPK (adenosine monophosphate protein kinase) activation, mTOR (protein kinase complex mechanistic target of rapamycin) complex 1 inhibition or microglia and astrocyte activation in the spinal dorsal horn [18,19].

Currently, the treatment for neuropathy is only successful for a minority of patients, with less than half experiencing positive results. Additionally, the medications prescribed according to current guidelines often lead to severe side effects [20]. Moreover, it was reported that there was a decrease in the efficiency of drugs from all therapeutic classes, which consequently led to an increase in the number of patients needing to be treated [21]. As a consequence, combination therapy is considered to be preferred among clinicians for alleviating pain from DN [22,23,24]. This rationale is based on two theories: (1) a phenotypically guided treatment improves symptomatic control (it is suggested that different clinical signs and symptoms may give clues to the mechanisms that induced the pain) and offers the possibility of a treatment approach based on the physiopathology of pain; and (2) targeting multiple mechanisms involved in neuropathy by administering a combination of active substances from different therapeutic classes, which is considered to be superior to monotherapy [25,26]. All things considered, combined therapy may be a better and more efficient option for reducing pain from DN.

The primary objective of the current research was to reinforce and to expand on the findings of our previous study, where we showed that the combination of sildenafil and metformin had pain-relieving effects in a mouse model of DN [4], considering the following aspects: (1) rats are a reliable model for studying diabetes and its complications, closely mimicking human disease responses to environmental agents like toxins, stress, and diet, and their consistent reactions to alloxan facilitate reproducible results [27,28,29,30]; (2) rat physiology is easier to track and has accumulated valuable information for studying the metabolic profile and pathology of various stages of type 2 diabetes [31]; (3) rats possess larger and more developed nervous system structures compared to mice, providing detailed information on neuropathic changes and yielding data that more accurately approximate human responses [32]; (4) rats require shorter behavior habituation periods, are easier to train in conditioning tasks, and experience less stress induced by experimenters and procedures compared to mice [33]; (5) rats are preferred over mice in certain behavioral assays due to their complex and discernible behavioral patterns, aiding in the observation and interpretation of neuropathic effects and treatments [34]; and (6) using both mouse and rat models enhances research findings’ robustness and generalizability, strengthening evidence and supporting experimental conclusions. In this study, we evaluated how various combinations of sildenafil and metformin affected pain sensitivity in a rat model of DN, as well as their impact on certain biochemical markers such as TNF-α, IL-6, nitrites, and thiols levels. 

## 2. Results

To enhance the readability and comprehension of this text, preventing unnecessary word repetition, we used acronyms for the experimental groups included in this study, as described in Table 1. Gabapentin was used as the positive control, considering it is recommended as a first-line therapy in neuropathic pain [35].

### 2.1. Blood Glucose Levels

Following the administration of alloxan to induce diabetes, alterations in blood glucose levels were detected when comparing the ND group to the diabetic groups. This comparison was conducted using a univariate analysis of variance (ANOVA), yielding a significant F value of 32.09 (*p* < 0.0001). The results are illustrated in Figure 1A. The results of our study demonstrated statistically significant elevations in the measured values across all groups with diabetes as compared to the non-diabetic group (*p* < 0.05, Figure 1A).

Significant variations in blood glucose levels across groups were seen after 7 days of therapy (univariate ANOVA, F = 48.89, *p* < 0.0001, Figure 1B). All diabetic groups maintained a high level of blood glucose, except the S2.5 + M300 and S3 + M500 groups, which recorded significant decreases in blood glucose when compared to the D group (*p* < 0.01; *p* < 0.001, Figure 1B).

At the end of the experiment, the blood glucose concentration was significantly changed following the treatment (univariate ANOVA, F = 51.87, *p* < 0.0001, Figure 1C). We noticed significant decreases for all three groups treated with sildenafil–metformin combinations (*p* < 0.001, Figure 1C). Moreover, the groups S2.5 + M300 and S3 + M500 demonstrated the best glycemic control, with percentages of lowering blood glucose levels of 62.53% and 65.35%, respectively, compared to the initial values (Figure 1D).

### 2.2. Tests for the Evaluation of Antihyperalgesic Effect

#### 2.2.1. Heat Hypersensitivity

The findings from our study indicate variations in pain reaction latency observed in the hot plate test when comparing the ND group to all diabetes groups before medication administration (univariate ANOVA, F = 19.98, *p* < 0.0056, Figure 2A). The diabetes groups had statistically reduced pain response values in comparison to the ND group (*p* < 0.01, Figure 2A).

After 7 days of treatment, we noted changes in the pain reaction latency (univariate ANOVA, F = 21.69, *p* < 0.0029, Figure 2B). The G90 and G150 groups recorded significant increases in pain reaction latency when compared to the D group (*p* < 0.01, *p* < 0.001, Figure 2B). Group G150 demonstrated the most pronounced antihyperalgesic effect, with a pain sensitivity reduction effect of 97.52%. Regarding the groups treated with the sildenafil–metformin combinations, the S2.5 + M300 and S3 + M500 groups showed significant decreases in pain sensitivity when compared to the D group, with effects of 88.9% and 92.24%, respectively (*p* < 0.01, Figure 2B).

After administering the treatments for 14 consecutive days, our research revealed changes regarding the pain perception parameter (univariate ANOVA, F = 27.85, *p* < 0.0002, Figure 2C). Administration of gabapentin at a dose of 150 mg · kg^−1^ produced the strongest antihyperalgesic effect of 97.52%, followed by gabapentin at a dose of 90 mg · kg^−1^ with an effect of increasing pain reaction latency by 78.4% versus the D group (*p* < 0.001, *p* < 0.05, Figure 2C). The S2.5 + M300 and S3 + M500 groups demonstrated significant decreases in pain sensitivity when compared to the D group, with comparable effects to the G150 group (*p* < 0.01, Figure 2C).

#### 2.2.2. Cold Hypersensitivity

The cold plate test at −5 °C showed differences between the ND group and the diabetic groups (univariate ANOVA, F = 22.92, *p* < 0.0018, Figure 2D), with increased pain sensitivity in the diabetic groups when compared to the ND group (*p* < 0.05, Figure 2D).

The administration of treatment produced variations among the groups, both after 8 days (univariate ANOVA, F = 23.5, *p* < 0.0014, Figure 2E) and after 15 days of treatment (univariate ANOVA, F = 30.88, *p* < 0.0001, Figure 2F). Thus, among the groups treated with gabapentin, only the G90 and G150 groups recorded significant increases in pain reaction latency versus the D group after 7 days of treatment (*p* < 0.001, *p* < 0.05, Figure 2E). At the end of the experiment, the highest antihyperalgesic effect was demonstrated by the G150 group (280.15%), followed by the G90 group (228.35%).

Among the sildenafil–metformin combinations, both after 8 days and after 15 days of treatment, only the S2.5 + M300 and S3 + M500 groups significantly reduced pain sensitivity when compared to the D group (*p* < 0.05, Figure 2E,F). The most striking effect was shown by the S3 + M500 group in the two moments of determination (177.1%, respectively 267.78%), followed by the S2.5 + M300 group, with an effect of reducing pain sensitivity of 145.57% and 165.72%, respectively, when compared to the D group. In addition, the S3 + M500 group demonstrated increasing effects of pain reaction latency comparable to those of the G150 group after 15 days of treatment (267.78% vs. 280.15%).

#### 2.2.3. Tactile Hypersensitivity

After inducing diabetes, an initial test was performed before administering the therapy, and variances among the groups were noticed (ANOVA, F = 2.26, *p* < 0.039, Figure 3A). Our research revealed that the diabetic groups showed significant increases in pain sensitivity when compared to the ND group (*p* < 0.05, Figure 3A).

Following the administration of the treatment, changes were detected after 7 days (ANOVA, F = 20.61, *p* < 0.0044, Figure 3B), as well as at the end of the experiment (ANOVA, F = 18.67, *p* < 0.0093, Figure 3C). Among the groups treated with gabapentin, only the group that received the dose of 150 mg · kg^−1^ recorded a significant decrease in tactile sensitivity when compared to the D group after 7 days of treatment (*p* < 0.05, Figure 3B). A comparable effect was demonstrated by the S3 + M500 group (*p* < 0.05, Figure 3B), with a pain sensitivity reduction effect of 63.99%, compared to 66.88% in the G150 group.

After 14 days of treatment, the G90 and G150 groups demonstrated significant increases in the 50% withdrawal threshold in the von Frey test when compared to the D group (*p* < 0.01, Figure 3C). The most pronounced antihyperalgesic effect was demonstrated by the group treated with a dose of 150 mg · kg^−1^ of gabapentin (79.18%). On the other hand, our research highlighted the fact that the S2.5 + M300 and S3 + M500 combinations significantly reduced pain sensitivity when compared to the D group, with effects of 71.18% and 63.25%, respectively (*p* < 0.05, Figure 3C).

### 2.3. Biochemical Assay of Rat Brain and Liver Homogenates

#### 2.3.1. Assessment of TNF-α and Il-6

Considering the cytotoxic effects of alloxan-induced diabetes on multiple organs, such as the brain, pancreas, liver, and kidney [36], we measured the levels of pro-inflammatory cytokines TNF-α and IL-6 in the brain and liver tissues after the experiment was finished. Furthermore, our aim was to investigate whether the combination of sildenafil and metformin could efficiently reduce the concentration of these cytokines in tissues.

Regarding the level of TNF-α, the ANOVA test revealed significant variations among the groups, both in the brain tissues (F = 4.156, *p* = 0.0018, Figure 4A) and in the liver tissues (F = 26.15, *p* < 0.0001, Figure 4B). The D group and the G30 group showed significant increases when compared to the ND group, while all the other diabetic groups reduced the level of TNF-α, with significantly lower values for the G90 and S3 + M500 groups in the brain tissue when compared to the D group (*p* < 0.05, Figure 4A), with a percentage of 75.39% and 94.4%, respectively. On the other hand, in liver tissues, TNF-α concentration was markedly increased in the D group, while the G90, G150, S2.5 + M300, and S3 + M500 groups had significantly reduced levels when compared to the D group (*p* < 0.05, Figure 4B).

Following the biochemical analysis of brain tissue samples (ANOVA, F = 9.553, *p* < 0.0001, Figure 4C) and liver tissue samples (ANOVA, F = 27.88, Figure 4D), significant variations in IL-6 levels were observed after 15 days of treatment. Thus, in both brain and liver tissues, we noted significant increases for the D group, G30 group, and S2 + M100 group when compared to the ND group (*p* < 0.05, Figure 4C,D). IL-6 production was significantly decreased following treatment with the dose of 150 mg · kg^−1^ of gabapentin and the S2.5 + M300 and S3 + M500 combinations when compared to the D group in brain tissues (*p* < 0.05, Figure 4C). On the other hand, in liver tissues, the administration of gabapentin in doses of 90 mg · kg^−1^ and 150 mg · kg^−1^, and of S2.5 + M300 and S3 + M500 combinations significantly reduced the level of IL-6 when compared to the D group (*p* < 0.05, Figure 4D), with decreases of 67.08%, 72.29%, 39.95%, and 57.15%, respectively.

#### 2.3.2. Assessment of NOS Activity

Besides pro-inflammatory cytokines, we also evaluated the concentration of nitrites and total nitrites as end products of NOS in brain and liver homogenates, seeing that iNOS (inducible nitric oxide synthase) produces cellular toxicity in many systems [37].

The unifactorial ANOVA test revealed significant differences between groups in terms of nitrites’ concentration in the brain tissues (F = 8.667, *p* < 0.0001, Figure 5A) and the liver tissues (F = 7.496, *p* < 0.0001, Figure 5B), but also regarding the concentration of total nitrites both in the brain tissues (F = 3.562, *p* = 0.0045, Figure 5C) and in the liver tissues (F = 7.912, *p* < 0.0001, Figure 5D).

Regarding the concentration of nitrites, the D group recorded significant increases when compared to the ND group both in the brain and in the liver tissues (*p* < 0.001, Figure 5C,D). All diabetic groups that received treatment markedly reduced the concentration of nitrites in the brain tissues (*p* < 0.05, Figure 5A), while in the liver tissues, except the G150 group, all treated diabetic groups demonstrated significant decreases in nitrites when compared to the D group (*p* < 0.05, Figure 5B).

In brain tissues, the concentration of total nitrites was significantly increased for the D group when compared to the ND group, while all other diabetic groups had significantly decreased total nitrites’ concentrations when compared to the D group (*p* < 0.05, Figure 5C). In liver tissues, the D group, G30 group, and S2 + M100 combination recorded significantly increased levels when compared to the ND group (*p* < 0.05, Figure 5D). The concentration of total nitrites was markedly reduced for the G90 and G150 groups, but also for the S2.5 + M300 combination, with decreases of 60.87%, 38.13%, and 31.92%.

#### 2.3.3. Assessment of Total Thiols

After 15 days of treatment, the concentration of total thiols was also evaluated in the brain and liver tissues.

The biochemical analysis showed significant variations in the concentration of total thiols among the groups both in the brain tissues (univariate ANOVA, F = 3.535, *p* < 0.0053, Figure 5E) and in the liver tissues (univariate ANOVA, F = 4.517, *p* = 0.053, Figure 5F). The D group showed significant decreases in the concentration of total thiols reported as glutathione equivalents in both liver and brain tissues (*p* < 0.05, Figure 5E,F). Among the groups that received treatment, all sildenafil–metformin combinations increased the concentration of total thiols, but only for the S2.5 + M300 group was this increase statistically significant and only in liver tissues when compared to the D group, with an increasing percentage of 136.57% (*p* < 0.05, Figure 5F).

Among the groups treated with different doses of gabapentin, only the G150 group produced significant increases in both brain and liver tissues when compared to the D group (*p* < 0.01, Figure 5E,F).

## 3. Discussions

The present research confirms our previous findings that the sildenafil–metformin combination effectively reverses thermal hypersensitivity [4]. Additionally, our current research demonstrates that this combination can also reduce mechanical hypersensitivity following a 14-day treatment period in alloxan-induced DN in rats. Multi-drug therapy could be a more efficient option than single-drug therapy because combinations of two drugs could reduce the frequency of side effects by lowering the dose of each drug [25]. Moreover, sildenafil and metformin act through different mechanisms to reduce pain from neuropathy, which could be essential in a disease characterized by complex pathophysiology such as DN.

Given that gabapentin is recommended as a first-line therapy for neuropathic pain, particularly in post-herpetic neuralgia and DN [35], we used it as the positive control in our study. This allows us to compare the efficacy of the novel treatment against a clinically established benchmark, ensuring meaningful assessment and prioritizing animal welfare. Furthermore, testing gabapentin at three different doses provides a thorough understanding of its dose–response relationship within this study’s context and aids in comparing the dose–response effects of the sildenafil–metformin combinations. Our current study confirms that gabapentin efficiently reverses thermal and mechanical hyperalgesia in a dose-dependent manner in hot plate, cold plate, and von Frey tests in alloxan-induced DN in rats.

Many preclinical and clinical investigations have demonstrated over time that systemic inflammation plays a role in the etiology of diabetic peripheral neuropathy [38]. TNF-α signaling exacerbates vascular inflammation and oxidative stress in type 2 diabetes, contributing to DN development [39]. Regarding IL-6, it has been most consistently associated with DN, affecting glial cells and neurons [40,41].

Various animal models of DN offer different perspectives, but challenges arise when comparing rodent models to human DN due to differences in variables such as diabetes type, induction method, and severity of symptoms [42,43,44]. In this study, we used a single 130 mg/kg^−1^ dose of alloxan to induce DN in rats, avoiding the cumulative toxicity of higher doses used in mice (three consecutive doses of 150 mg · kg^−1^) in our previous research [4]. Alloxan-induced hyperglycemia leads to neuropathic changes like reduced nerve conduction velocity, increased sensitivity to heat and cold, mechanical sensitivity, delayed gastrointestinal motility, and oxidative stress [44,45]. This method is cost-effective, easily replicable, and produces a high percentage of diabetic animals [46]. Consistent with previous studies, we observed that high blood sugar levels increase pain sensitivity, confirmed by our results showing increased pain sensitivity in the D group.

Metformin has been widely recognized for decades as a first-line therapy for managing blood sugar levels in patients with type 2 diabetes, as indicated by the current guidelines [47]. Its ability to improve insulin sensitivity and reduce blood glucose levels can influence metabolic processes that impact nerve function and pain perception [48]. Sildenafil’s mechanism of action involves increasing blood flow through vasodilation, potentially affecting peripheral blood flow and conditions with a vascular component [49]. This vasodilatory effect could be further enhanced by metformin’s ability to improve endothelial function [50]. Furthermore, both sildenafil and metformin exhibit anti-inflammatory effects—sildenafil through its impact on nitric oxide and cGMP and by inhibiting NF-κB (nuclear factor kappa B) and MAPKs (mitogen-activated protein kinases) [51,52,53,54,55], and metformin through AMPK activation and subsequent anti-inflammatory pathways [56,57,58]. On the other hand, preclinical research showed that the cold hypersensitivity test revealed synergism between sildenafil and metformin, with sildenafil enhancing metformin’s potency and vice versa. This indicates a reduced required dose of each drug when administered together, compared to when they are administered alone, for reducing heat hyperalgesia in DN-induced mice [59]. Additionally, metformin has been found to possess neuroprotective properties that are beneficial in conditions involving neuropathic pain [60]. Sildenafil’s vasodilatory effects may also contribute to improved nerve function and reduced ischemic damage [17,61,62]. Since DN involves both metabolic and vascular factors causing nerve damage and pain [63], the combination of sildenafil and metformin presents a promising approach. Considering that this therapy could improve blood flow and reduce inflammation, we found that the combination could alleviate neuropathy, a major diabetes complication.

In the current research, we expanded our analysis of biochemical markers to include the levels of total nitrites and total thiols in brain and liver homogenates, in addition to proinflammatory cytokines and nitrites. Regarding the concentration of total nitrites, in brain tissues, all three sildenafil–metformin combinations markedly reduced the level. In addition, all the groups treated with gabapentin significantly decreased the concentration of the total nitrites in brain tissues. On the other hand, the total thiols/protein ratio increased following the administration of sildenafil–metformin combinations and gabapentin. Thiols are organic compounds with a sulfhydryl (-SH) group. Our study measured total thiols as glutathione (GSH) equivalents. GSH stores cysteine, which is extracellularly sensitive and can auto-oxidize to cystine, generating harmful oxygen free radicals, impacting diabetes’ progression and complications [13,64]. Both types of diabetes show decreased protein thiol levels due to metabolic and inflammatory changes [14]. Previous studies indicated that metformin and sildenafil protect against low thiol content [65,66,67], and gabapentin also reverses low thiol concentrations in a neuropathic pain animal model [68].

One limitation of our study is the relatively short duration of administration of the drugs (15 days) due to the high toxicity of alloxan. Consequently, we were unable to examine the impact of sildenafil–metformin on biochemical markers in the context of long-term pathology. The second limitation may be regarded as the heightened fluctuation in pain sensitivity [69].

Overall, our study demonstrated, for the first time, to the best of our knowledge, that the sildenafil–metformin combination could efficiently alleviate alloxan-induced DN in rats. Our results reinforce the observations made in other studies according to which a combination of two drugs could be more efficient in treating the symptoms of DN [70]. This also aligns with our earlier research on the sildenafil–metformin combination, which showed promising results in reversing thermal hyperalgesia and reducing the levels of nitrites in mice [4]. Furthermore, our current study reveals that the combination treatment can also reverse mechanical hyperalgesia, increase total thiols’ levels in the liver and brain tissues, and decrease the production of inflammatory markers TNF-α and Il-6, as well as total nitrites’ concentrations.

## 4. Materials and Methods

### 4.1. Experimental Animals

This research was conducted on male Wistar rats (n = 110) aged 8–10 weeks from INCDMI Cantacuzino (Cantacuzino National Institute of Research, Bucharest, Romania). The rats were kept in Plexiglass cages and were given unlimited access to food and water. This research was conducted under controlled conditions, with the temperature maintained between 21 and 24 °C, and the humidity levels kept between 45% and 60%, monitored using a hygrothermometer.

This experiment was conducted according to the bioethical regulations governing research on experimental animals as outlined in Directive 2010/63/EU of the European Parliament and Law 43/2014, which has been subsequently amended and expanded by Law 199/2018. The experimental protocol (CFF07/10.04.2023) was authorized by the Bioethics Committee of the Faculty of Pharmacy, Carol Davila University of Medicine and Pharmacy, Bucharest, Romania.

### 4.2. Induction of Diabetes Mellitus and Treatments

Diabetes was induced by alloxan (Sigma Aldrich, Hamburg, Germany, product no. A7413) injection i.p. (intraperitoneally) in a single dose of 130 mg · kg^−1^ to rats (320 ± 10 g) that were kept fasting for 24 h prior. Considering that previous research indicated that the percentage of diabetic animals following the administration of alloxan is about 66%, we used 100 animals for the induction of diabetes [69]. After 48 h following alloxan administration, we determined the blood sugar of the animals with an ACCU-CHEK Active glucometer (Roche Diagnostic, Penzberg, Germany) by collecting blood from the veins of the tail (puncture) [69]. Out of 100 animals, 70 of them became hyperglycemic (blood glucose level > 180 mg/dL) and were divided into equal groups (n = 10) for the experimental tests. Besides these, we included a group of 10 non-diabetic rats in the non-diabetic control group. The substances were administered daily for 15 days by oral gavage as follows: ND group—non-diabetic control group that received distilled water 1 mL · kg^−1^; D group—diabetic control group that received distilled water 1 mL · kg^−1^; G30 group—gabapentin 30 mg · kg^−1^; G90 group—gabapentin 90 mg · kg^−1^; G150 group—gabapentin 150 mg · kg^−1^; S2 + M100 group—sildenafil 2 mg · kg^−1^ + metformin 100 mg · kg^−1^; S2.5 + M300 group—sildenafil 2.5 mg · kg^−1^ + metformin 300 mg · kg^−1^; and S3 + M500 group—sildenafil 3 mg · kg^−1^ + metformin 500 mg · kg^−1^. Sildenafil was procured from Actavis Group PTC EHF (Hafnarfirdi, Island), metformin from Gedeon Richter PLC (Budapest, Hungary), and gabapentin from Egis Pharmaceuticals PLC (Budapest, Hungary).

### 4.3. Blood Glucose Levels

Blood glucose levels (measured in mg/dL) were assessed at three-time points: initially, 7, and 15 days after the formation of the experimental groups. Blood samples were collected from the tail veins (puncture).

### 4.4. Tests for the Evaluation of Antihyperalgesic Effect

#### 4.4.1. Heat Hypersensitivity

The evaluation of heat hypersensitivity was conducted using the hot plate test, both at the beginning and after 7 and 14 days of treatment. The experimental subjects, in this case rats, were subjected to a thermal stimulus by being placed on a hot plate with a temperature of 53 °C. The time interval between the initiation of the thermal stimulus and the onset of the first observable indication of pain (licking/shaking of paws or jumping) was recorded as the latency period [71].

#### 4.4.2. Cold Hypersensitivity

For the evaluation of cold hypersensitivity, we used the cold plate test. The first manifestation of pain (licking/shaking of paws or jumping) was recorded for the rats placed on a plate cooled to −5 °C. The evaluation was determined initially and after 8 and 15 days of treatment [72,73].

For both cold and heat hypersensitivity, a maximum of 25 s of latency was maintained to avoid possible tissue damage.

#### 4.4.3. Tactile Hypersensitivity

Tactile hypersensitivity was assessed using von Frey filaments (Ugo Basile, Italy) initially and after 7 and 14 days of treatment. Animals were allowed to acclimate for 30 min in individual Plexiglas cages positioned on top of a perforated wire platform. Afterwards, von Frey filaments with increasing stiffness were used (1.4; 2; 4; 6; 8; 10; 15; and 26 g corresponding to the sizes 4.17; 4.31; 4.56; 4.74; 4.93; 5.07; 5.18; and 5.46) and applied with a moderate pressure to allow the filament to bend slightly for 6 sec. The filaments were applied to the 2 hind paws on their plantar surfaces. The choice of filaments was made so that the filament with the highest resistance was less than or equal to 10% of the weight of the rats. Using filament number 4 in the series and a force of 6 g, the test was started. The filament with the following stiffness was utilized if the animal did not remove its paw, which was regarded as a negative response (denoted O). The filament with the lesser stiffness was employed if the animal removed its paw, which was recorded as a positive reaction and marked with an X. Following the receipt of an OX or XO series, or four consecutive positive or negative responses, a total of four responses were conducted. Dixon’s method [74] was applied and validated by Chaplan et al. [75], and was used to calculate the 50% withdrawal threshold.

### 4.5. Biochemical Assay of Rat Brain and Liver Homogenates

For sacrifice, the animals were administered a dosage of 200 mg · kg^−1^ of thiopental sodium (Sigma Aldrich, St. Louis, MO, USA) [33]. Subsequently, the brains and livers were collected for subsequent analysis. Tissue homogenates were prepared by homogenizing tissue and phosphate-buffered saline (PBS) at a ratio of 1:10 (*w*/*v*) using a RW 14 basic homogenizer (IKA, Staufen, Germany). Then, the homogenates were diluted 1:10 with PBS before being subjected to experimental methods.

#### 4.5.1. Assessment of TNF-α and Il-6

The levels of TNF-α (catalog no. LS-F2558-1) and Il-6 (catalog no. LS-F5113-1) were evaluated by the instructions provided in the manual guide (LifeSpan BioSciences, Inc., Seattle, WA, USA).

#### 4.5.2. Assessment of NOS Activity

Utilizing the previously adjusted Griess method [76] (which reduced nitrate to nitrites using vanadium (III) [77], we were able to identify nitrites and total nitrites as NOS end products in the liver and brain. After five minutes at room temperature, the tissue homogenate (50 μL) was mixed with 100 μL of modified Griess reagent (Sigma Aldrich, St. Louis, MO, USA). The optical density (OD) was then measured at 540 nm. After measuring a NaNO standard curve (Sigma Aldrich, St. Louis, MO, USA), the protein content was expressed as NO_2_ equivalents (μM). For every sample, a blank was read, and any necessary adjustments were performed.

#### 4.5.3. Assessment of Total Thiols

A previously described method [78] using a sample to Ellman reagent ratio of 1:4 was used to determine total thiols. The results are expressed as the ratio of glutathione (GSH) equivalents (pM) to protein content (mg/mL).

#### 4.5.4. Protein Content

The widely recognized Lowry method was utilized to assess the total protein content in tissue homogenates and mitochondrial preparations [79].

### 4.6. Statistical Analysis

The statistical analysis of the results was conducted using GraphPad Prism v.5.00 software developed by GraphPad Software in San Diego, CA, USA. The distribution of the data was evaluated using the D’Agostino–Pearson test. For normally distributed data, one-way analysis of variance (ANOVA) test and Dunnett’s post hoc test were employed for analysis. In cases of non-parametric data, the Kruskal–Wallis test and Dunn’s post hoc test were utilized. It is considered that the observed differences between the groups are statistically significant when the value of *p* < 0.05, and the experimental results were expressed as individual mean values ± standard error of the mean (S.E.M.). We used Formula (1) to calculate the percentage differences in the experimental outcomes between the groups [80]:Δ% = (Mx − My)/My × 100(1)
where Mx is the mean value for D when compared to ND, or G30, G90, G150, S2 + M100, S2.5 + M300 and S3 + M500 groups when compared to D; and My is the mean value for either ND or D.

## 5. Conclusions

In conclusion, our study found that when inducing DN by a single dose of alloxan, the combination of sildenafil and metformin enhanced pain reaction latency in the von Frey test, reduced inflammation by lowering TNF-α and IL-6 levels, and protected against glutathione depletion, besides reversing thermal hyperalgesia and decreasing the concentrations of nitrites.

Considering all these, the treatment of DN with a combination of sildenafil and metformin may be a better option than those recommended by current guidelines.

## Figures and Tables

**Figure 1 pharmaceuticals-17-00783-f001:**
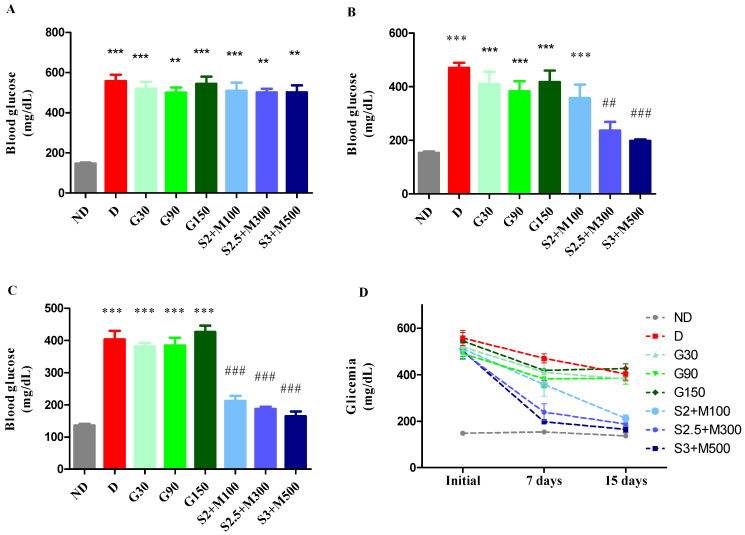
(**A**) Initial glycemia. (**B**) Glycemia at 7 days. (**C**) Glycemia at 15 days. (**D**) Evolution of the mean blood glucose levels during the experiment. Values are expressed as mean + S.E.M (standard error of the mean). ** *p* < 0.01; *** *p* < 0.001 vs. ND. ^##^
*p* < 0.01; ^###^
*p* < 0.001 vs. D.

**Figure 2 pharmaceuticals-17-00783-f002:**
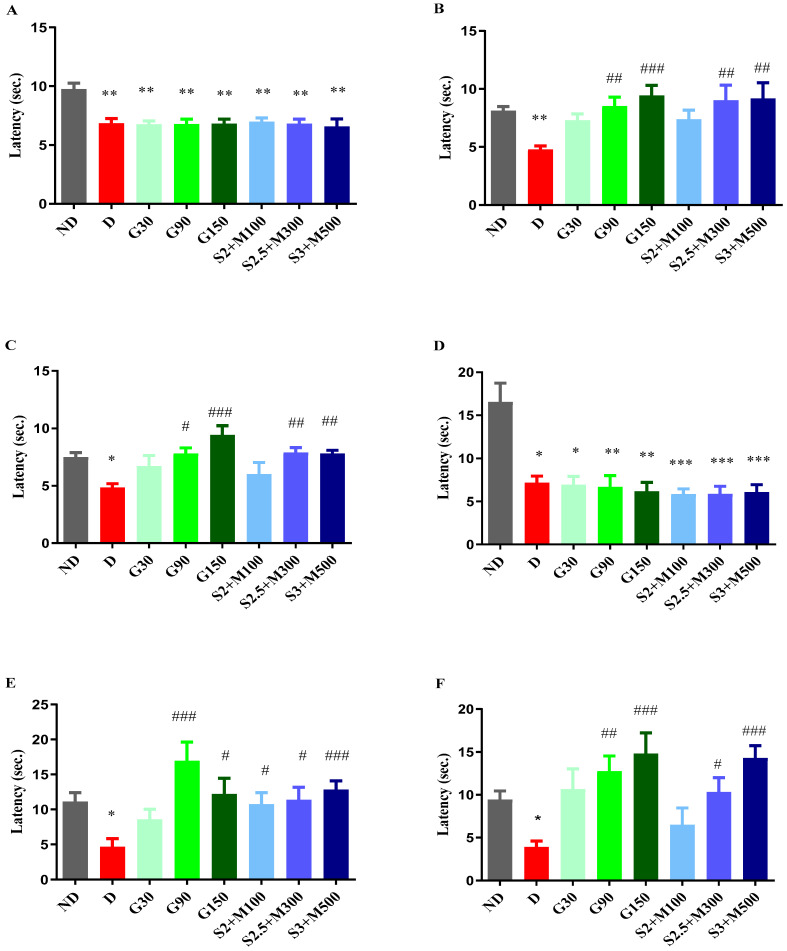
(**A**) Hot plate test—initial pain reaction latency. (**B**) Hot plate test—pain reaction latency after 7 days. (**C**) Hot plate test—pain reaction latency after 14 days. (**D**) Cold plate test—initial pain reaction latency. (**E**) Cold plate test—pain reaction latency after 8 days. (**F**) Cold plate test—pain reaction latency after 15 days. Values are expressed as mean + S.E.M. * *p* < 0.05; ** *p* < 0.01; *** *p* < 0.001 vs. ND. ^#^
*p* < 0.05; ^##^
*p* < 0.01; ^###^
*p* < 0.001 vs. D.

**Figure 3 pharmaceuticals-17-00783-f003:**
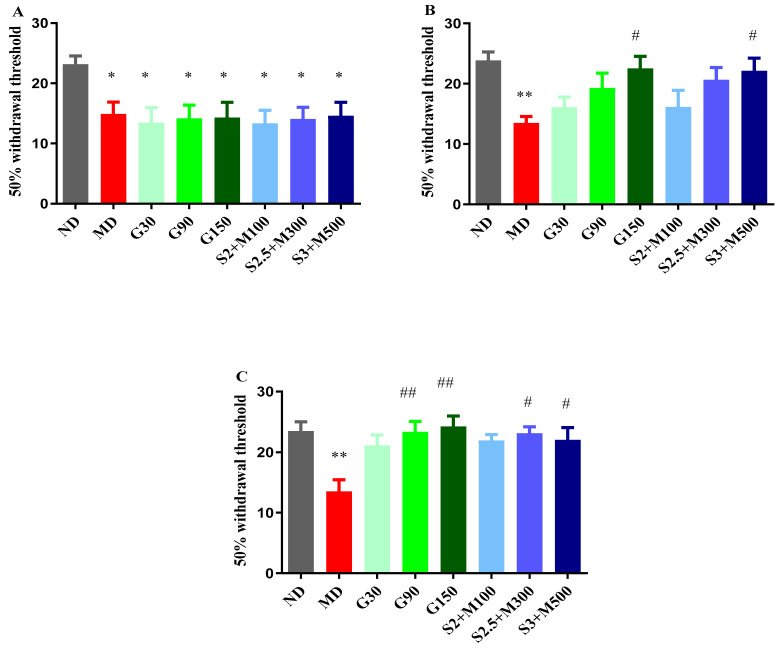
(**A**) Withdrawal threshold of 50% in initial von Frey test. (**B**) Withdrawal threshold of 50% in von Frey test at 7 days. (**C**) Withdrawal threshold of 50% in von Frey test at 14 days. Values are expressed as mean + S.E.M. * *p* < 0.05; ** *p* < 0.01 vs. ND. ^#^
*p* < 0.05; ^##^
*p* < 0.01 vs. D.

**Figure 4 pharmaceuticals-17-00783-f004:**
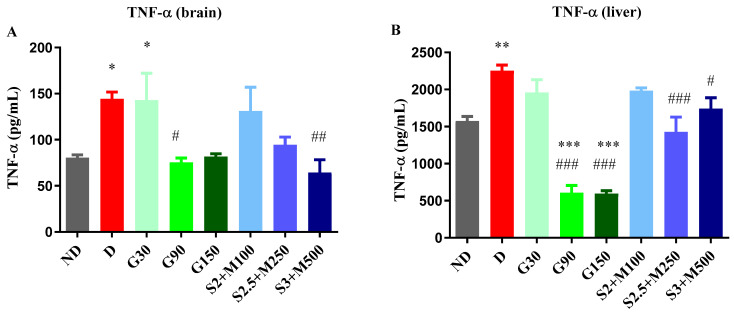
(**A**) Variation in TNF-α in rat brain tissues. (**B**) Variation TNF-α in rat liver tissues. (**C**) Variation in IL-6 in rat brain tissues. (**D**) Variation in IL-6 in rat liver tissues. Values are expressed as mean + S.E.M. * *p* < 0.05; ** *p* < 0.01; *** *p* < 0.001 vs. ND. ^#^
*p* < 0.05; ^##^
*p* < 0.01; ^###^
*p* < 0.001 vs. D.

**Figure 5 pharmaceuticals-17-00783-f005:**
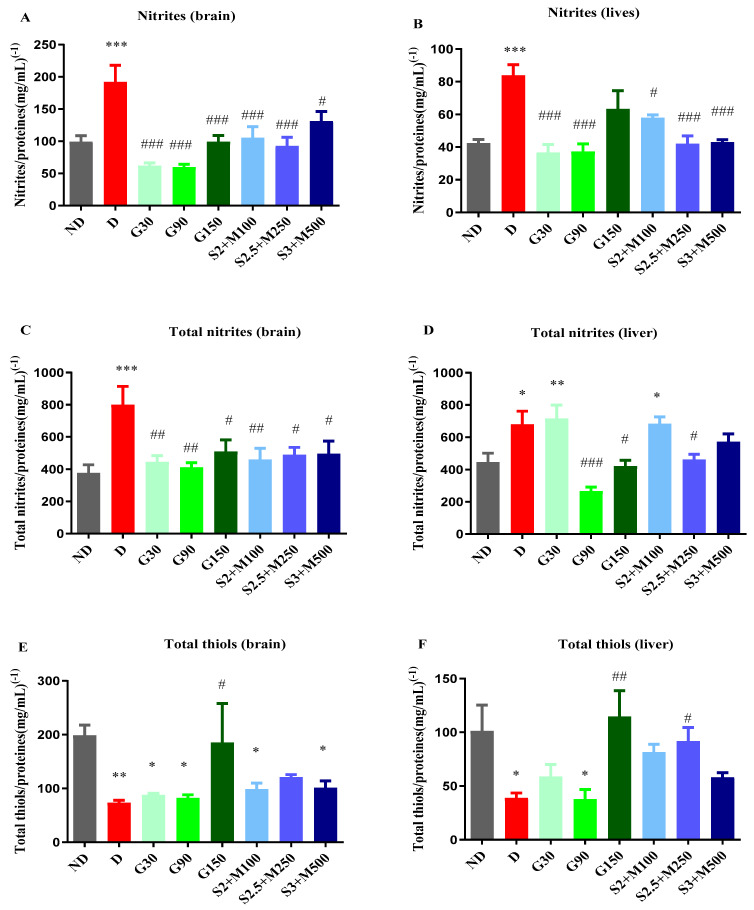
(**A**) Variation in nitrites/proteins ratio in rat brain tissues. (**B**) Variation in nitrites/proteins ratio in rat liver tissues. (**C**) Variation in total nitrites/proteins ratio in rat brain tissues. (**D**) Variation in total nitrites/proteins ratio in rat liver tissues. (**E**) Variation in total thiols/proteins ratio in rat brain tissues. (**F**) Variation in total thiols/proteins ratio in rat liver tissues. Values are expressed as mean + S.E.M. * *p* < 0.05; ** *p* < 0.01; *** *p* < 0.001 vs. ND. ^#^
*p* < 0.05; ^##^
*p* < 0.01; ^###^
*p* < 0.001 vs. D.

**Table 1 pharmaceuticals-17-00783-t001:** The experimental groups included in the present study and their acronyms.

Experimental Group	Acronyms
Non-diabetic control	**ND**
Diabetic control	**D**
Gabapentin 30 mg · kg^−1^	**G30**
Gabapentin 90 mg · kg^−1^	**G90**
Gabapentin 150 mg · kg^−1^	**G150**
Sildenafil 2 mg · kg^−1^ + Metformin 100 mg · kg^−1^	**S2 + M100**
Sildenafil 2.5 mg · kg^−1^ + Metformin 300 mg · kg^−1^	**S2.5 + M300**
Sildenafil 3 mg · kg^−1^ + Metformin 500 mg · kg^−1^	**S3 + M500**

## Data Availability

All data generated or analyzed during this study are included in this published article.

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
