# Peer review of "Evaluating the Antihyperalgesic Potential of Sildenafil–Metformin Combination and Its Impact on Biochemical Markers in Alloxan-Induced Diabetic Neuropathy in Rats"

_pharmaceuticals, 2024, doi:10.3390/ph17060783_

Round 1

Reviewer 1 Report

Comments and Suggestions for Authors

After reading the manuscript and the previous works of the authors, I got the impression that the authors presented the results of related experiments in a short time. In 2023, the authors published at least three papers on the effects of metformin, sildenafil and their combinations on pain and biochemical  indicators in mice. Now the authors have presented very similar results using rats. It remains unclear what is the fundamental novelty of this study and its contribution to solving the problem of diabetic neuropathy. What is the significant difference between the mouse model and the rat model, because  the authors did not use anything new (using rats in this study): the same behavioral and biochemical tests, the same drug combinations. If rats are the best model for these studies, then this should be justified in the Introduction. It should be reflected why, after the mouse model, the authors switched to rats - especially since the same results were obtained. Nothing new.

The authors write in the Conclusions section that "The purpose of our study was to investigate if the combination of sildenafil and metformin acts synergistically in reducing pain in alloxan-induced DN in rats, but also to elaborate on the results of our previous research in which we induced DN in mice." but this purpose was not declarated in the Introduction.

The Discussion section is written at great length. And some of its provisions look very "strained". For example, the thesis that a combination of these drugs is used for erectile dysfunction in men with diabetes, so this combination needs to be tested in a pain model. What is the interactions between them?

Next, the authors discuss the results of numerous studies using the alloxan model, which again raises the question of the novelty of the data they obtained and their scientific significance.

 In any case, the design of the presented experiment looks questionable. The authors set out to investigate the effect of a combination of two well-known drugs on a number of indicators related to pain and inflammation. However, they did not choose each of the drugs separately for comparison, but a completely different drug. Why Gabapentin at three various doses was used but the authors did not discuss this?

Moreover, based on the previous publications of the authors, it is quite possible that some of the data obtained has now been excluded in order to use them further. Isn't there an artificial division of results here to increase publication activity and self-citation?

Thus, I believe that this manuscript cannot be published in its present form. Serious work is needed to make the manuscript clear, logical and meaningful.

Author Response

Dear referee,

We hope this letter finds you well.

We deeply appreciate your valuable and insightful feedback. We are confident that the revisions have greatly enhanced the manuscript. Your time and intellectual contributions to reviewing our research are sincerely appreciated. We have endeavored to address all your comments in the revised manuscript to the best of our ability, and you can track the changes in the resubmitted version.

Point-by-point response to Reviewer’s comments and suggestions

1) After reading the manuscript and the previous works of the authors, I got the impression that the authors presented the results of related experiments in a short time. In 2023, the authors published at least three papers on the effects of metformin, sildenafil and their combinations on pain and biochemical indicators in mice. Now the authors have presented very similar results using rats. It remains unclear what is the fundamental novelty of this study and its contribution to solving the problem of diabetic neuropathy. What is the significant difference between the mouse model and the rat model, because the authors did not use anything new (using rats in this study): the same behavioral and biochemical tests, the same drug combinations. If rats are the best model for these studies, then this should be justified in the Introduction. It should be reflected why, after the mouse model, the authors switched to rats - especially since the same results were obtained. Nothing new.

Response: We justified in the Introduction why, after the mouse model, we decided to evaluate the sildenafil-metformin combination in a rat model, highlighting the differences between mice and rats models. Although we evaluated the same combination of drugs as in our previous mouse model, there are several differences between the studies:

  1. We induced diabetic DN in rats by administering a single, lower dose of alloxan (130 mg/kg) compared to the three doses used in mice (150 mg/kg). This approach helped avoid cumulative toxicity and demonstrated the rats' higher sensitivity to alloxan, leading to more pronounced DN symptoms. The animal models used in the study differ in the administered alloxan dose and the number of doses to obtain diabetic animals for research. In mice, the cumulative toxicity of three doses of alloxan caused significant changes in cytokine release (IL6 and TNFα) in the liver and brain, which were not always significantly reduced by the administered combinations. The use of rats in experimental pharmacology stems from the fact that the physiology of the rat more closely resembles the corresponding human condition (the rat is more intelligent than the mouse and is capable of learning a wider variety of tasks, including modifying the response to pain under conditions of exposure to temperature stimuli). Literature data have shown that in diabetes models, the rat model behaves more similarly to the human disease, including in terms of the response to environmental agents ( toxins, stress, diet) [1].

[1]       P. M. Iannaccone and H. J. Jacob, “Rats!,” Dis. Model. Mech., vol. 2, no. 5–6, pp. 206–210, Apr. 2009, doi: 10.1242/dmm.002733.

  1. In addition to assessing thermal sensitivity, we also evaluated tactile hypersensitivity in the rats. We used a different cold test (cold-plate test instead of tail withdrawal test). The experimental tests used in this study contribute to the validation of the experimental model, as well as to the identification of the sites and mechanisms of action of the sildenafil-metformin combination, as follows:
  2. a) The hot plate test allows for the investigation of the effects of the test substances on A fibers (A-MH type II, which innervate hairy skin) that respond to temperatures slightly lower than the pain perception threshold [2].
  3. b) The cold plate test allows for the identification of the involvement of fibers and sensitive nociceptors (non-nociceptive afferents sensitive to cold are spontaneously active at normal skin temperature, and their excitability increases as the temperature decreases) [3].

[2]       S. N. Raja, R. A. Meyer, and J. N. Campbell, “Peripheral Mechanisms of Somatic Pain,” Anesthesiology, vol. 68, no. 4, pp. 571–590, Apr. 1988, doi: 10.1097/00000542-198804000-00016.

[3]       M. Koltzenburg, C. L. Stucky, and G. R. Lewin, “Receptive Properties of Mouse Sensory Neurons Innervating Hairy Skin,” J. Neurophysiol., vol. 78, no. 4, pp. 1841–1850, Oct. 1997, doi: 10.1152/jn.1997.78.4.1841.

  1. We expanded our analysis of biochemical markers to include total nitrites and total thiols levels in brain and liver homogenates, in addition to proinflammatory cytokines and nitrites.All these biochemical determinations complement the antihyperalgesic profile and contribute to understanding the mechanisms of action of the sildenafil/metformin combination in alloxan-induced diabetic rats.

  1. The dosages used in this study (sildenafil-2; 2.5; 3 mg/kg and metformin 100; 300; 500 mg/kg) were determined based on our previous results conducted in mice (using dose-response pharmacological relationships). Additionally, previous research allowed for the use of gabapentin (a reference substance recommended by guidelines) in three progressive doses: 30; 90; 150 mg/kg to establish a correct dose-effect relationship for anti-hyperalgesic effect.

Overall, these differences in experimental design and analysis contribute to a more comprehensive understanding of the potential benefits of sildenafil-metformin combination in treating DN.

2) The authors write in the Conclusions section that "The purpose of our study was to investigate if the combination of sildenafil and metformin acts synergistically in reducing pain in alloxan-induced DN in rats, but also to elaborate on the results of our previous research in which we induced DN in mice." but this purpose was not declared in the Introduction.

Response: We rewrite the main purpose of the study so that it is the same in the Introduction and in the Conclusion.

3) The Discussion section is written at great length. And some of its provisions look very "strained". For example, the thesis that a combination of these drugs is used for erectile dysfunction in men with diabetes, so this combination needs to be tested in a pain model. What is the interactions between them?

Response: We shortened the Discussion section, highlighting the differences between the previous study on mice and the current one on rats. Moreover, we explained why we chose to test the combination between sildenafil and metformin in DN.

4) Next, the authors discuss the results of numerous studies using the alloxan model, which again raises the question of the novelty of the data they obtained and their scientific significance.

Response: We explained the advantages of using the alloxan DN induction model and why we chose this method. Recent literature discusses the importance of various animal models used to investigate diabetes and its complications. This is because diabetes is an endocrine disease with complex pathophysiology regarding both the disease itself and its complications. Since animal models are essential for preliminary research on this disease, it becomes necessary to describe the advantages that an experimental model brings to such research [4]. However, the novelty of our study does not refer to the alloxan induction model, but the evaluation of a new possible treatment in DN.

[4]       D. N. Athmuri and P. A. Shiekh, “Experimental diabetic animal models to study diabetes and diabetic complications,” MethodsX, vol. 11, p. 102474, Dec. 2023, doi: 10.1016/j.mex.2023.102474.

5) In any case, the design of the presented experiment looks questionable. The authors set out to investigate the effect of a combination of two well-known drugs on a number of indicators related to pain and inflammation. However, they did not choose each of the drugs separately for comparison, but a completely different drug. Why Gabapentin at three various doses was used but the authors did not discuss this?

Response: The experimental design of the study is clear and falls within the domain of drug repurposing research for two medications targeting the core treatment of the disease, hyperglycemia (metformin), and one of its complications, ED (sildenafil).

The study methodology is clearly presented through the following aspects: the animal model, the tested combinations, the doses used in combination in rats (obtained from our previous determinations in mice), the pharmacological methods for investigating neuropathic pain (internationally accepted), and the reference substance, which should be a substance included in therapy guidelines as the first choice (gabapentin). The two substances administered separately (metformin or sildenafil) cannot be used as reference substances, as they are not currently authorized as antihyperalgesic medications.

Moreover, we explained in the Discussion section why we chose to use gabapentin as the reference substance and why we used 3 different doses of it. For the two experimental animals, the doses of sildenafil [5][6][7] and metformin [8][9][10][11][12][13][14][15] used in the experiments are similar. In previous research, we observed that administering these drugs as monotherapies produced positive results on neuropathic pain. For bioethical reasons, we did not consider it relevant to compare the combinations with the substances administered in monotherapy for the experiment. The main purpose of our research was to expand on the results obtained on mice considering the sildenafil-metformin combination, because combination therapy is considered to be preferred among clinicians for alleviating pain from DN.

[5]       L. Wang et al., “Sildenafil Ameliorates Long Term Peripheral Neuropathy in Type II Diabetic Mice,” PLoS One, vol. 10, no. 2, p. e0118134, Feb. 2015, doi: 10.1371/journal.pone.0118134.

[6]       C. PuÈ™caÈ™u 1 et al., “INVESTIGATION OF ANTIHYPERALGESIC EFFECTS OF DIFFERENT DOSES OF SILDENAFIL AND METFORMIN IN ALLOXAN-INDUCED DIABETIC NEUROPATHY IN MICE,” Farmacia, vol. 71, p. 3, 2023, doi: 10.31925/farmacia.2023.3.19.

[7]       N. K. Jain, C. . Patil, A. Singh, and S. K. Kulkarni, “Sildenafil-induced peripheral analgesia and activation of the nitric oxide–cyclic GMP pathway,” Brain Res., vol. 909, no. 1–2, pp. 170–178, Aug. 2001, doi: 10.1016/S0006-8993(01)02673-7.

[8]       Y. Liu et al., “AMP-Activated Protein Kinase Activation in Dorsal Root Ganglion Suppresses mTOR/p70S6K Signaling and Alleviates Painful Radiculopathies in Lumbar Disc Herniation Rat Model,” Spine (Phila. Pa. 1976)., vol. 44, no. 15, pp. E865–E872, Aug. 2019, doi: 10.1097/BRS.0000000000003005.

[9]       V. Das, J. S. Kroin, M. Moric, R. J. McCarthy, and A. Buvanendran, “AMP-activated protein kinase (AMPK) activator drugs reduce mechanical allodynia in a mouse model of low back pain,” Reg. Anesth. Pain Med., vol. 44, no. 11, pp. 1010–1014, Nov. 2019, doi: 10.1136/rapm-2019-100839.

[10]     J. Ma, H. Yu, J. Liu, Y. Chen, Q. Wang, and L. Xiang, “Metformin attenuates hyperalgesia and allodynia in rats with painful diabetic neuropathy induced by streptozotocin,” Eur. J. Pharmacol., vol. 764, pp. 599–606, Oct. 2015, doi: 10.1016/J.EJPHAR.2015.06.010.

[11]     X. J. Cao et al., “Metformin attenuates diabetic neuropathic pain via AMPK/NF-κB signaling pathway in dorsal root ganglion of diabetic rats,” Brain Res., vol. 1772, p. 147663, Dec. 2021, doi: 10.1016/J.BRAINRES.2021.147663.

[12]     Q. Huang, Y. Chen, N. Gong, and Y.-X. Wang, “Methylglyoxal mediates streptozotocin-induced diabetic neuropathic pain via activation of the peripheral TRPA1 and Nav1.8 channels,” Metabolism, vol. 65, no. 4, pp. 463–474, Apr. 2016, doi: 10.1016/j.metabol.2015.12.002.

[13]     S. Wang et al., “Negative Regulation of TRPA1 by AMPK in Primary Sensory Neurons as a Potential Mechanism of Painful Diabetic Neuropathy,” Diabetes, vol. 67, no. 1, pp. 98–109, Jan. 2018, doi: 10.2337/db17-0503.

[14]     Q.-L. Mao-Ying et al., “The Anti-Diabetic Drug Metformin Protects against Chemotherapy-Induced Peripheral Neuropathy in a Mouse Model,” PLoS One, vol. 9, no. 6, p. e100701, Jun. 2014, doi: 10.1371/journal.pone.0100701.

[15]     P. S. A. Augusto et al., “Metformin antinociceptive effect in models of nociceptive and neuropathic pain is partially mediated by activation of opioidergic mechanisms,” Eur. J. Pharmacol., vol. 858, p. 172497, Sep. 2019, doi: 10.1016/j.ejphar.2019.172497.

6) Moreover, based on the previous publications of the authors, it is quite possible that some of the data obtained has now been excluded in order to use them further. Isn't there an artificial division of results here to increase publication activity and self-citation?

Response: All pharmacological testing and biochemical evaluations reported in the protocol authorized by the Bioethics Committee of the Faculty of Pharmacy, Carol Davila University of Medicine and Pharmacy, Bucharest, Romania (CFF07/10.04.2023) were included in this study. Regarding the self-citation, we cited only 2 study conducted by us in this research, as this is not one of our interest-the self citation. Citing the two previous studies conducted in mice is necessary, on one hand, to justify the use of sildenafil and metformin doses in diabetic rats, and on the other hand, to demonstrate the synergistic action regarding the antihyperalgesic effect of the combinations.

Finally, we sincerely believe that your suggestions have substantially improved the manuscript, and given us a better scientific experience for future studies. Thank you for your valuable input.

With kindest regards,

The authors

Reviewer 2 Report

Comments and Suggestions for Authors The article is well-written and structured, with a clear introduction that explains the underlying mechanisms of diabetic neuropathy in an effective way. The introduction is also supported by up-to-date literature. The research goals and the gap in current literature are clearly stated too. The results are well-explained and the discussions are coherent with the findings. The tables and figures are particularly effective in conveying the information.   However, it would be beneficial to consider adding a dedicated paragraph to fully explain the materials and methods section (lines 18-21: "Methods: The study com- 18 prised a cohort of 70 diabetic rats and 10 non-diabetic rats which were submitted to hot, cold, and 19 tactile stimulus tests. Moreover, we investigated the influence of this combination on TNF-α, IL-6, 20 nitrites and thiols levels in brain and liver tissues). This would provide a clearer understanding of the study's methodology.   Additionally, there are some minor issues with the use of English terminology, such as using "At present" instead of "At the present time" (line 13) and "new" instead of "novel" (line 14). Comments on the Quality of English Language There are some minor issues with the use of English terminology, such as using "At present" instead of "At the present time" (line 13) and "new" instead of "novel" (line 14).

Author Response

Dear referee,

We hope this letter finds you well.

We deeply appreciate your valuable and insightful feedback. We are confident that the revisions have greatly enhanced the manuscript. Your time and intellectual contributions to reviewing our research are sincerely appreciated. We have endeavored to address your comments in the revised manuscript to the best of our ability, and you can track the changes in the resubmitted version.

Taking your suggestions into consideration, we have added a paragraph in the abstract to the Materials section to fully explain the study methodology. However, the fully explained study methodology can be find at the Materials and Methods section in the manuscript.

Additionally, we have reviewed the English terminology once again.

Thank you for your valuable input.

With kindest regards,

The authors

Round 2

Reviewer 1 Report

Comments and Suggestions for Authors

The authors have worked on the text, but I still cannot recommend the manuscript for publication. Yes, I have no questions about the experimental part (although the experimental design does not seem entirely good to me), but the quality of the manuscript itself clearly needs improvement. I ask authors to understand that the quality of a manuscript affects its reception and, ultimately, relevance, and to carefully consider my recommendations.

Please, change the Lines 93-110 (The primary objective of the current research....) with Lines 91-93 (In this study, we evaluated how...) for the aim of the study was the last in the Introduction.

The discussion is still written unclearly. All the work talks about pain and neuroinflammation, and here erectile dysfunction comes up in the first sentence. For what? Why?

Please start the discussion with lines 393-413 (tThe present research confirms....). Then Lines 420- 431, Lines 339-349, after that Lines 292-315. in Lines 314-315 changed the "aimed" with "found". Then Lines 471- 

The use of gabapentin as a control(?) should be specified in section 2 (Lines 113-115) and discussed in the Discussion.

What about Conclusions...In my opinion there is no need to duplicate the purpose of the study. In conclusions, please focus on the conclusions.

Author Response

Dear Referee,

Thank you once again for your suggestions, which will undoubtedly enhance the quality of our manuscript.

Point-by-point response to Reviewer’s comments and suggestions

Please, change the Lines 93-110 (The primary objective of the current research....) with Lines 91-93 (In this study, we evaluated how...) for the aim of the study was the last in the Introduction.

Response: We made all the changes you suggested in the Introduction.

The discussion is still written unclearly. All the work talks about pain and neuro-inflammation, and here erectile dysfunction comes up in the first sentence. For what? Why? Please start the discussion with lines 393-413 (The present research confirms....). Then Lines 420- 431, Lines 339-349, after that Lines 292-315. In Lines 314-315 changed the "aimed" with "found". Then Lines 471- 

Response: We rewrote the discussions exactly according to the suggestions given.

The use of gabapentin as a control (?) should be specified in section 2 (Lines 113-115) and discussed in the Discussion.

Response: We specified the use of gabapentin as the positive control at the beginning of Results section.

What about Conclusions...In my opinion there is no need to duplicate the purpose of the study. In conclusions, please focus on the conclusions.

Response: We rewrote the conclusion without duplicating the study's purpose.

Finally, we genuinely believe that your suggestions have once again significantly improved the quality of the manuscript.

With kindest regards,

The authors

Round 3

Reviewer 1 Report

Comments and Suggestions for Authors

The manuscript can be published.